# A Comprehensive Approach for the Diagnosis of Primary Ciliary Dyskinesia—Experiences from the First 100 Patients of the PCD-UNIBE Diagnostic Center

**DOI:** 10.3390/diagnostics11091540

**Published:** 2021-08-25

**Authors:** Loretta Müller, Sibel T. Savas, Stefan A. Tschanz, Andrea Stokes, Anaïs Escher, Mirjam Nussbaumer, Marina Bullo, Claudia E. Kuehni, Sylvain Blanchon, Andreas Jung, Nicolas Regamey, Beat Haenni, Martin Schneiter, Jonas Ingold, Elisabeth Kieninger, Carmen Casaulta, Philipp Latzin

**Affiliations:** 1Division of Paediatric Respiratory Medicine and Allergology, Department of Paediatrics, Inselspital, Bern University Hospital, University of Bern, 3010 Bern, Switzerland; sibel.t.savas@med.uni-giessen.de (S.T.S.); Andrea.stokes@insel.ch (A.S.); anaisestelle.escher@insel.ch (A.E.); mirjam.nussbaumer@gmx.ch (M.N.); marina.bullo@insel.ch (M.B.); claudia.kuehni@ispm.unibe.ch (C.E.K.); elisabeth.kieninger@insel.ch (E.K.); carmen.casaulta@insel.ch (C.C.); philipp.latzin@insel.ch (P.L.); 2Department of BioMedical Research (DBMR), University of Bern, 3008 Bern, Switzerland; 3Institute of Anatomy, University of Bern, 3012 Bern, Switzerland; beat.haenni@ana.unibe.ch (B.H.); martin.schneiter@iap.unibe.ch (M.S.); j.ingold@gmx.ch (J.I.); 4Institute of Social and Preventive Medicine, University of Bern, 3012 Bern, Switzerland; 5Pediatric Pulmonology and Cystic Fibrosis Unit, Service of Pediatrics, Department Woman–Mother–Child, Lausanne University Hospital, University of Lausanne, 1011 Lausanne, Switzerland; Sylvain.Blanchon@chuv.ch; 6Division of Respiratory Medicine, University Children’s Hospital Zurich, 8032 Zurich, Switzerland; andreas.jung@kispi.uzh.ch; 7Division of Paediatric Pulmonology, Children’s Hospital Lucerne, 6000 Lucerne, Switzerland; nicolas.regamey@luks.ch; 8Institute of Applied Physics, University of Bern, 3012 Bern, Switzerland

**Keywords:** airways, ciliopathy, air-liquid interface cell culture, high-speed videomicroscopy, immunofluorescence, transmission electron microscopy

## Abstract

Primary ciliary dyskinesia (PCD) is a rare genetic disease characterized by dyskinetic cilia. Respiratory symptoms usually start at birth. The lack of diagnostic gold standard tests is challenging, as PCD diagnostics requires different methods with high expertise. We founded PCD-UNIBE as the first comprehensive PCD diagnostic center in Switzerland. Our diagnostic approach includes nasal brushing and cell culture with analysis of ciliary motility via high-speed-videomicroscopy (HSVM) and immunofluorescence labeling (IF) of structural proteins. Selected patients undergo electron microscopy (TEM) of ciliary ultrastructure and genetics. We report here on the first 100 patients assessed by PCD-UNIBE. All patients received HSVM fresh, IF, and cell culture (success rate of 90%). We repeated the HSVM with cell cultures and conducted TEM in 30 patients and genetics in 31 patients. Results from cell cultures were much clearer compared to fresh samples. For 80 patients, we found no evidence of PCD, 17 were diagnosed with PCD, two remained inconclusive, and one case is ongoing. HSVM was diagnostic in 12, IF in 14, TEM in five and genetics in 11 cases. None of the methods was able to diagnose all 17 PCD cases, highlighting that a comprehensive approach is essential for an accurate diagnosis of PCD.

## 1. Introduction

Primary ciliary dyskinesia (PCD) is a rare genetic disorder (prevalence between 1:10,000 and 1:20,000 [1,2,3]) that manifests with chronic respiratory symptoms caused by impaired mucociliary clearance [4,5,6]. Symptoms include chronic wet cough, chronic rhinitis, recurrent otitis media, and subsequent hearing impairment, recurrent respiratory infections, and may start early in life with neonatal respiratory distress [6,7,8,9,10]. Pulmonary long-term morbidity includes function impairment and structural changes (e.g., lung atelectasis, bronchiectasis) [8,11,12,13]. The impaired ciliary function with inefficient or absent beating is based on defects in a large number of genes. Those genes primarily involve the ciliary motor protein family, the dyneins, but also other components of the ciliary structure [9,14]. Despite recent diagnostic guidelines of the European Respiratory Society (ERS) [15] and the American Thoracic Society (ATS) [16], the diagnosis of PCD remains challenging. Difficulties are related to several issues: (i) clinical presentation shows a wide phenotypical spectrum and is not specific for PCD; (ii) there is no single test that is diagnostic for PCD as a stand-alone test; current diagnostic guidelines therefore include a combination of different functional, structural and molecular methods; (iii) all diagnostic methods require a high level of expertise; (iv) over ¼ of the genetic defects causing PCD are unknown [17]; (v) analytical methods have not been standardized sufficiently [18] and have unsatisfactory sensitivity and/or specificity if considered individually; and (vi) PCD is a rare disease and satisfactory diagnostic experience is available in only a few specialized centers. Hence, PCD is often diagnosed late in life or not at all, resulting in an overall underdiagnosis and undertreatment [1,2]. Even if causal therapy is not yet available, early diagnosis and interdisciplinary management increases quality of life and prognosis of affected patients [19,20].

The current diagnostic guidelines include a combination of different functional, structural, and molecular methods. All require a high level of expertise. Currently, the following methods are used routinely for the diagnosis of PCD: (i) measurement of nasal nitric oxide (nNO); (ii) high-speed-videomicroscopy (HSVM) to assess ciliary motility of viable cells (fresh samples or air-liquid interface (ALI) cell cultures); (iii) immunofluorescence staining (IF) of different structural proteins; (iv) assessment of the axonemal ultrastructure by transmission electron microscopy (TEM); and (v) genetic testing for the identification of pathogenic or likely pathogenic variants in PCD-associated genes.

Based on 30 years of experience in structural PCD diagnosis, we founded the interdisciplinary center for comprehensive PCD diagnostics (PCD-UNIBE) at the University Children’s Hospital, Inselspital, Bern University Hospital and the Institute of Anatomy, University of Bern, in January 2018. This is the first comprehensive PCD diagnostic center in Switzerland and is led by an experienced team of clinicians and biomedical researchers.

The aim of this study is to summarize the experiences of the first 100 patients assessed by PCD-UNIBE and to report on the workflow, benefits, and challenges of our center and the diagnostic algorithm used by us.

## 2. Materials and Methods

### 2.1. Study Design and Study Population

This observational study includes the first 100 patients referred to our PCD-UNIBE diagnostic center. The study was approved by the ethics committees of the University Children’s Hospital Bern (Ethikkommission der Kinderkliniken) and of the Canton Bern (Kantonale Ethikkomission Bern), Switzerland (project identification code 2018-02155, 01/2019). We obtained written informed consents from all participants or their legal guardians.

### 2.2. Diagnostic Workflow

The diagnostic workflow used at PCD-UNIBE implements the ERS guidelines [15], but adds (for all patients) ALI cell culture of brushed cells and IF of structural proteins of the ciliary axoneme. In brief, we perform HSVM and IF for all patients, but execute TEM and genetics analysis only for selected patients. Our diagnostic workflow is shown in Figure 1 and our detailed diagnostic algorithm has been recently published [21]. An important part of PCD-UNIBE are interdisciplinary meetings with clinicians (pneumologists) and diagnostic research specialists to discuss cases, decide on further investigations and confirm diagnosis. nNO measurements are not performed by PCD-UNIBE, but by the referring centers themselves (since some patients are not physically present at PCD-UNIBE, but only their nasal brushes are sent to us).

### 2.3. Nasal Brushing and Further Processing

Nasal epithelial cells (NECs) are obtained by brushings with adapted interdental brushes from both nostrils. The cells are then removed from the brushes and used for different investigations: (i) HSVM of the fresh sample; (ii) preparation of slides for IF staining; (iii) cultivation of cells; and (iv) fixation for TEM (if the sample contains sufficiently large groups of ciliated cells). Detailed descriptions of the nasal brushing itself, its processing and the further procedures are provided in the Appendix B—Supplementary Methods’ Description.

### 2.4. Cell Culture

Primary NECs are cultured using the PneumaCult Media Kits (Stemcell Technologies), according to the manufacturer’s protocol with minor changes. A detailed description of our protocol is available in Appendix B.

### 2.5. High-Speed Videomicroscopy (HSVM)

Ciliary motility of fresh or ALI cells is analyzed via HSVM. To summarize, we use a silicon spacer to build an imaging chamber and record videos on an inverted bright field microscope. These videos are then analyzed using our own analysis software (termed “Cilialyzer” [23]) by assessing the ciliary beating pattern (CBP), beating frequency (CBF), coordination of cilia movement, and particle transport. A detailed description of the methods is provided in Appendix B. The Excel analysis template is available online under the Appendix A.

### 2.6. Immunofluorescence Staining (IF)

Different structural proteins of the ciliary axoneme are labeled using standard protocols for IF. Our standard panel includes dynein axonemal heavy chain 5 (DNAH5), growth arrest specific 8 (GAS8), and radial-spoke-head 9 (RSPH9) (and dynein axonemal light intermediate chain 1 (DNALI1) for earlier samples). Based on findings from the IF and the HSVM, further proteins are stained. A detailed description of the methods and a list of available proteins are provided in Appendix B.

### 2.7. Transmission Electron Microscopy (TEM)

We perform an ultrastructural analysis of cilia via TEM for patients with high clinical suspicion and/or suspicious results from the standard tests (HSVM and IF). In short, cells (fresh or from ALI cultures) are fixed and processed, and 100–200 well assessable cilia cross sections imaged. The axonemal structures are systematically evaluated and scored according to the international consensus guidelines on TEM in PCD diagnosis [24]. Additionally, we discriminate between proximal and distal localization. A detailed method’s description is provided in Appendix B and the Excel analysis template is available online under the Appendix A.

### 2.8. Genetical Analysis

Analysis of pathogenic or likely pathogenic variants in all currently known PCD-associated genes is performed via next-generation sequencing of the whole exome by specialized centers; for more details see Appendix B.

## 3. Results

### 3.1. Study Population

The first 100 patients referred to the new PCD-UNIBE center for comprehensive PCD diagnostics came from various hospitals all over Switzerland, mostly children’s clinics. Details on the study population are presented in Table 1.

### 3.2. Overview of the Tests: HSVM, IF, TEM, Genetics

The diagnostic outcome was usually the result of several tests; see Figure 2 for an overview over all patients (A) and PCD cases (B). For all patients, we performed HSVM of the fresh samples and tried to obtain re-differentiated ALI cell cultures. HSVM ALI was performed whenever we succeeded in growing and re-differentiating cells (see below for details on cell culture success rate). For the vast majority of cases, we performed IF, preferable from ALI cell cultures (*n* = 82, *n* = 49 additional from fresh samples). For one case, we omitted IF since the clinical suspicion was moderate, HSVM showed normal CBP and CBF, and PCD was no longer the most probable diagnosis pursued. In 30 cases with highly suspicious CBP in HSVM and/or high clinical suspicion, we performed TEM. We carried out TEM analysis from seven fresh samples and from 26 ALI samples (for three patients we performed both). Genetics was done in 31 cases with high clinical suspicion and/or suggestive results of previous tests.

### 3.3. Air-Liquid Interface (ALI) Cell Culture

We aimed to cultivate cells from all patients. The cell culture grew and re-differentiated successfully for 90% from the first brushings. Reasons for failure of cell cultivation were bacterial contamination (*n* = 1), fungal contamination (*n* = 2), bad primary material (*n* = 7, e.g., not sufficient viable cells, no attachment or growing of the cells). In 10 cases, the cell cultivation was not successful at all; additionally, in one case, the quality of the cell culture was insufficient. For four patients, we omitted a re-brushing: one case could be diagnosed with PCD based on a TEM hallmark defect in the fresh sample; one patient had a pathologic mutation in a known PCD-associated gene; two patients refused a re-brushing. For the other seven cases with failure of cell cultivation, we performed a re-brushing and cultured cells again (success rate of 86%; six re-differentiated successfully, one failure due to fungal contamination).

### 3.4. Results of the Performed Tests: HSVM, IF, TEM, Genetics

From the 107 fresh HSVM analyses, 29 cases (27%) showed no evidence for PCD (CBP score > 3), 54 cases (50%) were inconclusive, and 24 cases (22%) showed a high evidence for PCD (Figure 3). Results of HSVM done on cell cultures were much clearer: 78 cases (83%) showed no evidence for PCD, four cases (4%) were inconclusive, and 12 cases (13%) showed high evidence for PCD. Interestingly, not all cases were assessed identically from fresh and ALI samples. For 42 cases, the conclusion was similar in both samples: 28 cases were assessed as “no evidence for PCD”, four cases as “inconclusive”, and 10 cases as “high evidence for PCD” based on both, fresh and ALI samples. Forty-one cases were assessed as “inconclusive” based on fresh samples and as “no evidence for PCD” based on the ALI sample. Moreover, importantly, one case was assessed as “no evidence for PCD” based on fresh and as “high evidence for PCD” based on ALI sample. For one case, the fresh sample was “inconclusive”, but the ALI sample showed a “high evidence for PCD”. In addition, nine cases showed “high evidence for PCD” in fresh and “no evidence for PCD” in the ALI samples. The samples of which the cell culture did not successfully re-differentiate were assessed as follows: 1× “no evidence for PCD”, 7× “inconclusive”, 5× “high evidence for PCD”. Figure 4A,B show the typical difference in sample quality for the assessment of the ciliary motility via HSVM.

The assessments of IF show similar characteristics, when comparing fresh vs. ALI, as the ones of HSVM (Figure 3): based on fresh samples statistically significant more inconclusive results (29 cases (55%) versus 19 cases (23%), *p* = 0.03) and fewer results suggesting “no evidence for PCD” (23 cases (43%) versus 53 cases (64%), *p* = 0.18) compared to the ALI samples. Furthermore, IF of the ALI samples enabled us more often to obtain a “high evidence for PCD” (one case (2%) versus 11 cases (13%), *p* = 0.38). Figure 4C–F give two examples of possible differences in the quality of fresh and ALI samples.

We performed TEM analysis for 30 cases (*n* = 4 only fresh, *n* = 3 fresh and ALI, *n* = 23 only ALI). There was “no evidence for PCD” in 22 cases (72%). Six cases were positive for PCD: four cases (14%) showed a class 1 ODA and IDA defect, one (3%) a class 1 microtubular disorganization and IDAs defect, and one (3%) a class 2 ODA defect. Two cases (7%) remained inconclusive due to bad quality of cell cultures. The production of TEM sections and images from ALI cell cultures was far more efficient than from freshly brushed samples. Abundant ciliated cells from large epithelial stripes and the possibility of aligning the cell culture prior to sectioning provided many more axonemal views, which are additionally optimally transected (see also Figure 4G,H). Furthermore, the effort needed for the TEM assessment of cell cultures was roughly one third of the one needed for the sparse, randomly oriented fresh brushing material.

Genetic analysis was carried out in 31 cases, with one analysis still ongoing. In 17 cases (55%), no abnormal variants were found. In nine cases (29%), two pathologic or likely pathologic variants in the same gene were found (7× DNAH11, 1× DNAH5, 1× HYDIN), finally leading to the diagnosis of PCD. Five cases (16%) showed genetic abnormalities, but did not lead to a (direct) PCD diagnosis: (1) in one case, a homozygous mutation of unknown significance in the DNAH11 gene was found (PCD was confirmed due to missing DNAH11 in IF and pathologic HSVM). (2) In another case, only a single likely pathogenic variant in the DNAH11 was found (PCD was confirmed due to missing DNAH11 in IF and pathologic HSVM). (3) In one case three variants of unknown significance in the DNAH5 gene were found (PCD was not confirmed in this case, since HSVM and TEM were normal and DNAH5 was present in IF). (4) In another case, we found two DNAH9 variants of unknown significance *in-cis* (the missing of DNAH9 in IF and a TEM class 2 ODA & IDA defect diagnosed PCD). (5) The last of the unclear cases showed only one pathogenic variant in the HYDIN gene (PCD was not diagnosed due to HSVM being better than expected for HYDIN mutations and normal nNO levels).

### 3.5. Overall Diagnostics Outcome

There was no evidence for PCD in 80 cases (Table 2). Seventeen patients were diagnosed with PCD (5× “high evidence for PCD”, 12× “PCD positive”). Three cases remained inconclusive: Two patients refused further investigations, and in one case, the diagnostics is still ongoing (The fresh sample and the cell culture showed only a few cilia. The genetic analysis showed no pathogenic or likely pathogenic variants in 43 genes (including MCIDAS, CCNO, FOXJ1). Thus, we performed a re-brushing of which the results were still pending). Among the 17 patients with diagnosed PCD, there are three cases with ongoing analysis: In one case, PCD was diagnosed based on fresh HSVM and TEM, but the genetic analysis is not yet complete. In two cases, PCD was diagnosed (1x based on HSVM and TEM, 1x based on IF), but the underlying genetic mutation could not be identified yet. Two of the cases with “no evidence for PCD” were clinically highly suspicious (one with repeatedly reduced nNO), but none of the tests (TEM, HSVM, IF, genetics) showed any evidence of PCD. Further investigations are planned for these patients.

### 3.6. Relevance of the Different Methods for the Diagnosis of PCD

The result of the HSVM confirmed the diagnosis of PCD in 13 cases (13% of all 100 cases investigated with this method). For IF, this was the case in 14 cases (14% of the 99 IF investigated cases), for TEM in five (17% of the 30 TEM investigated cases) and genetics in 10 cases (33% of the 30 genetically investigated cases) (Table 3). This shows that none of the methods used was able to diagnose all 17 PCD cases. Thus, a comprehensive approach is essential for an accurate diagnosis of PCD.

### 3.7. Costs

The costs for the different investigations vary a lot (Table 4) and are highly specific for different countries and healthcare systems. Our standard diagnostics including brushing, cell culture, HSVM fresh, and ALI and IF ALI costs EUR 1229. This is slightly more compared to the costs of a set without cell culture including three brushings, three HSVM fresh, and IF fresh, which adds up to EUR 1165. However, as soon as TEM needs to be performed, the costs for the cell culture are overcompensated: A full set with only fresh material (3× brushings, 3× HSVM fresh, IF fresh, TEM fresh) would cost EUR 3165, while a full set including cell culture (1x brushing, HSVM fresh and ALI, IF ALI, TEM ALI) would be EUR 2025. These reduced costs are in addition to the advantages of higher quality of the results (for details see above), less clinical visits and less burden for the patients due to fewer brushings. Our costs are higher compared to the numbers published earlier by Shoemark et al.: USD 187 (EUR 159 (exchange rate of EUR 0.85 per one dollar was used) and USD 1452 (EUR 1231). The higher costs in our setting are mostly related to higher salaries and higher prices for consumables in Switzerland.

## 4. Discussion

### 4.1. Summary

Among the first 100 patients referred to our newly founded PCD-UNIBE diagnostic center, we diagnosed PCD in 17 cases and found no evidence for PCD in 80 patients. Three cases remain inconclusive due to patients’ refusal for further investigations (*n* = 2) or ongoing diagnostics (*n* = 1). We performed HSVM of the fresh samples for all patients and IF for 99% of the patients. HSVM was repeated for all patients with a successful ALI cell culture (90% after the initial brushing, 5% after a re-brushing). TEM was performed for 30 and genetics for 31 patients. The use of the cell culture avoided a re-brushing in 90% of the cases and was crucial for most of the cases as it reduced the ratio of inconclusive findings for HSVM and IF significantly. HSVM confirmed diagnosis of PCD in 13, IF in 14, TEM in five, and genetics in 10 cases, respectively. None of the methods used was able to diagnose all 17 PCD cases.

### 4.2. Diagnostic Impact of Methods/Procedures in Our Setting

By setting up our new comprehensive PCD diagnostic center in 2018, we decided to include ALI cell culture as standard procedure for all patients. Herein, we report on the importance of this as well as of the other included methods. Every single method included in our algorithm was essential for certain cases to diagnose PCD. IF and genetics were the sole diagnostic method in one case each (in the case of genetics, the cell culture did not grow successfully, HSVM and IF of the fresh sample were inconclusive and we omitted a re-brushing since the genetic testing confirmed PCD). For all other cases, at least a combination of two of the four methods were needed to confirm the diagnosis of PCD. Nevertheless, since none of the methods was able to diagnose all cases, we can neither omit one of them nor choose a single method to be the best. Furthermore, we would like to emphasize the importance of ALI cell cultures for our diagnostic procedure. Its value has been described before [25,26,27,28], but we would like to add important aspects. The use of cell culture avoided a re-brushing in 90% of all cases. This is of particular importance for pediatric patients, in whom additional uncomfortable examinations should be avoided. Further results from HSVM, IF, and TEM using cell cultures are much more precise: The ratio of inconclusive assessments was significantly lower for HSVM and IF after cell culture compared to fresh samples. HSVM of cell cultures has already been shown to robustly represent original characteristics while eliminating secondary effects and overcoming low cell yields of the fresh samples [15,26,28,29]. Secondary or artifactual dyskinesia often seen in fresh samples is one of the main critical points raised when HSVM is used as a diagnostic method [17]. Even highly standardized and gentle brushings mostly provide merely tiny cell conglomerates or single cells that are ripped out of their natural environment. This may lead to additional analysis artefacts. Therefore, we principally avoid HSVM conclusions from single cells (as also recognized by international HSVM specialists). We also found a clear improvement of the sampling quality for IF and TEM as the amount of available cell material influences the validity of the diagnosis. While TEM from fresh material often requires time-consuming search of sparsely scattered cells in numerous resin blocks, ALI-derived epithelial stripes provide large lawns of trimmed cilia. This increases the chance of getting optimally oriented orthogonal transects in one or two sections from only two blocks. TEM assessment can thus usually be based on several hundreds of cilia cross sections obtained at half of the effort and costs needed for fresh samples. The same was true for IF: slides from cell culture contain more nicely stained epithelial cells reducing considerably the time needed to perform a valid analysis. An additional advantage of routinely performed cell cultivation is that leftover cells not used for diagnostics in the first round can be cryostored and used as backup cells (if the primary cell culture should not re-differentiate successfully or if further tests are needed (e.g., RNA sequencing from differentiated cells because of unclear genetics results)) and/or for research purposes (if patients consent), as recognized earlier [26].

### 4.3. Limitations and Strengths

Our study presents a realistic scenario using real world patients’ data. We principally highlight the practical aspects of our workflow, as such descriptions are often lacking in other studies. These could be particularly helpful for diagnostic centers. Our diagnostic algorithm includes an integral set of several methods including cell culture as preparatory step. Thus, we can evaluate the advantages and disadvantages of the different morpho-functional approaches HSVM, IF, TEM, and the potential improvement by the use of cell cultures. By our intention to present a real live report, patient data was exclusively obtained during our routine diagnostic workflow. A limitation of the study is that, according to our diagnostic workflow, TEM and genetics are not performed if previous results clearly showed no evidence for PCD and the clinical suspicion is low. In order to comply with state regulations to reduce healthcare costs, the expensive TEM analytics and genetic panel diagnostics were only performed upon high suspicion based on clinics and other test results. Additionally, genetics need a special approval by the health insurance. Therefore, we did not have results of all methods for all patients. Three cases remained inconclusive due to missing analytic data (2× refusal of further investigation, 1x ongoing analysis), but are consciously part of the first 100 patients referred to PCD-UNIBE. Such situations are an authentic part of the daily work in our PCD diagnostic center.

### 4.4. Comparison with the Literature

In our study population, prevalence of PCD diagnosis (17%) was rather high compared to a previous study with a prevalence of approximately 10% [22]. There are two reasons for this high diagnostic yield: (i) when we started the comprehensive approach for PCD diagnostic in 2018, we worked up several previously unclear, but clinically highly suspicious cases. The proportion of patients diagnosed with PCD among them was higher than among the routinely referred ones. This may falsely indicate that our comprehensive workflow enables a superior detection rate. (ii) Among the first 100 patients were two families with four and three members, respectively, also contributing to the high prevalence. The family with four members was affected by mutations in the *DNAH11* gene. Interestingly, we found many *DNAH11* cases (8 out of 10 genetically solved cases), also among the cases we work-up again. Additionally, one of the diagnosed, but genetically unsolved cases have one pathologic variant in the *DNAH11* gene (but so far, no second one). The high proportion of *DNAH11* mutations, especially in worked up cases, probably reflects the former diagnostic approach with focus on TEM (TEM hallmark defect is still officially needed to reclaim support from the Swiss disability insurance) and HSVM of fresh samples. The fact that cilia of patients with *DNAH11* mutations show a normal TEM ultrastructure explains that these cases had not been diagnosed previously. This highlights again that a comprehensive approach based on different methods is essential.

### 4.5. Clinical and Practical Relevance

Patients with suspicion of PCD have the right to know whether they suffer from PCD or not. Since every method on its own misses a considerable part of PCD cases (e.g., approximately 30% for TEM and genetics if tested individually [17]), all currently available methods are needed. In our study, every method missed at least four PCD cases (24% of all 17 PCD cases). Even though current therapy is similar for patients with a clinically high suspicion of PCD and those with confirmed PCD, it is still important to precisely know the underlying causes of clinical symptoms [30]. A clear diagnosis may be of importance for parents who are considering having more children. In addition, since some PCD mutations are associated with infertility, a clear diagnosis with known underlying genetic defect is important for the future family planning. Last, having a clear diagnosis of PCD and its specific genetic mutation is important with regard to therapeutic regimens. Recently, first evidence-based data for therapies, specifically for PCD (e.g., with macrolide antibiotics [31]), became available, and initiatives for molecular treatments or gene therapies for PCD are ongoing (e.g., CLEAN-PCD study, ClinicalTrials.gov Identifier: NCT02871778).

## 5. Conclusions

Diagnosis of PCD is challenging and every method included in our diagnostic algorithm contributed to final diagnoses. In our approach, the use of ALI cell cultures for HSVM, IF, and TEM was particularly crucial. However, none of the methods was able to diagnose all 17 PCD cases singlehandedly, highlighting that a comprehensive approach is essential for an accurate PCD diagnostics.

## Figures and Tables

**Figure 1 diagnostics-11-01540-f001:**
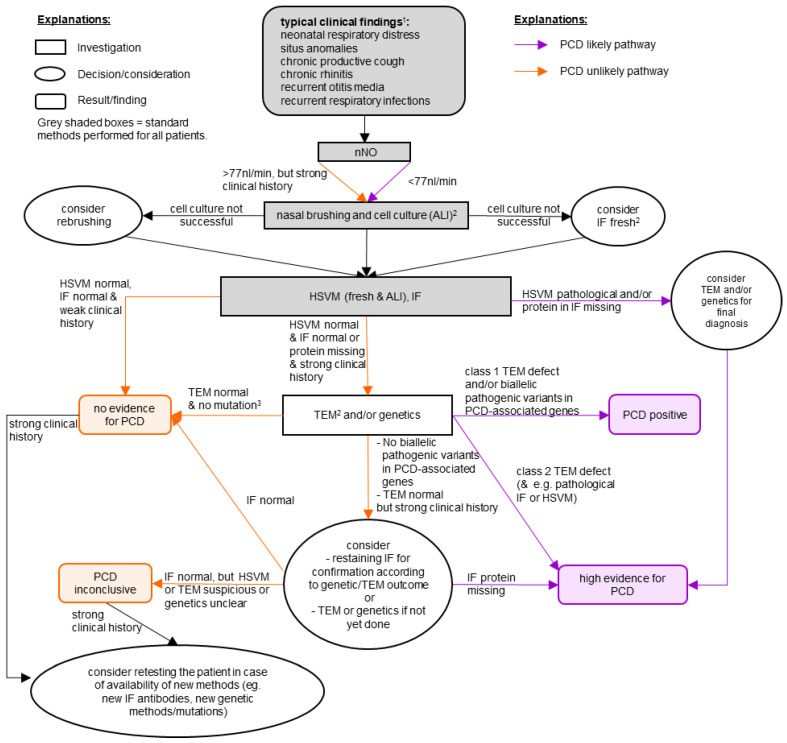
Diagnostic workflow of our comprehensive PCD-UNIBE diagnostic center. “PCD positive” and “high evidence for PCD” are considered as a diagnosis of PCD. For patients with a high clinical suspicion, we usually perform all available methods. ^1^ As a clinical screening, the PICADAR-Score [22] may be useful. ^2^ Further investigations (HSVM, IF, and TEM) are preferably conducted by analyzing the material of the ALI cell cultures. A re-brushing is considered if cell culture was not successful. When re-brushing was not possible, fresh material was used. ^3^ We recommend further investigations (e.g., RNA-analysis or array-CGH) if other results suggest PCD. Abbreviations: ALI: air-liquid interface, HSVM: high-speed-videomicroscopy, IF: immunofluorescence staining, nNO: nasal nitric oxide, PCD: primary ciliary dyskinesia, TEM: transmission electron microscopy.

**Figure 2 diagnostics-11-01540-f002:**
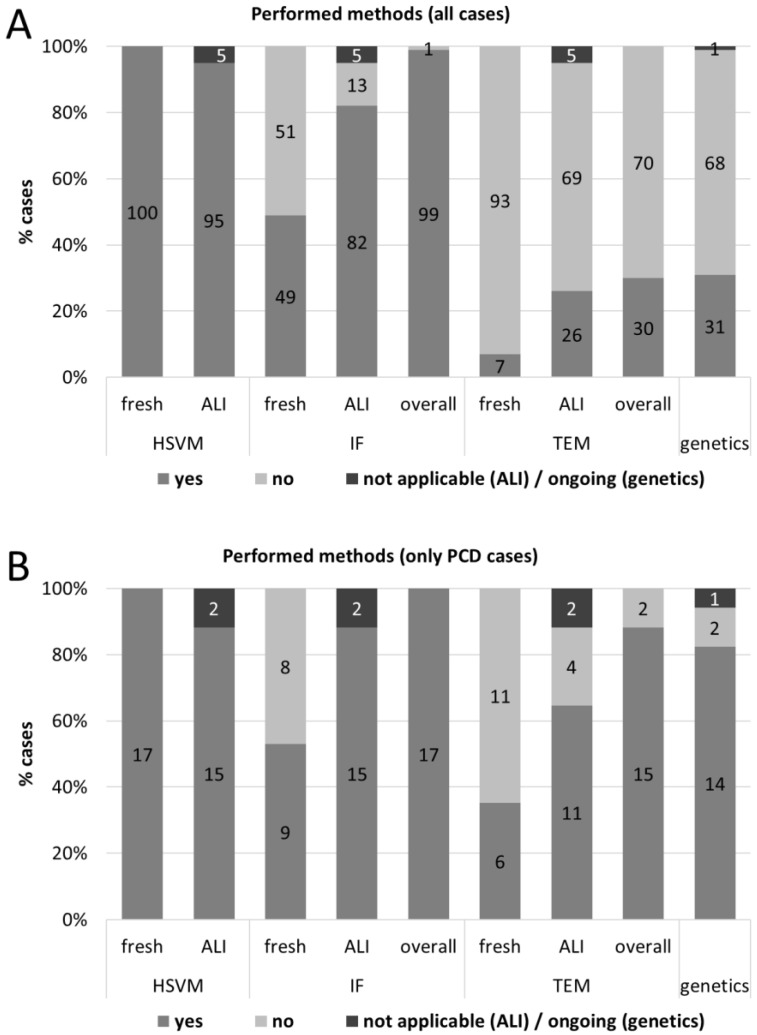
Overview of the performed tests from the first 100 patients (**A**) of the PCD-UNIBE diagnostic center and of PCD cases only (**B**). Numbers in the bars indicate number of cases.

**Figure 3 diagnostics-11-01540-f003:**
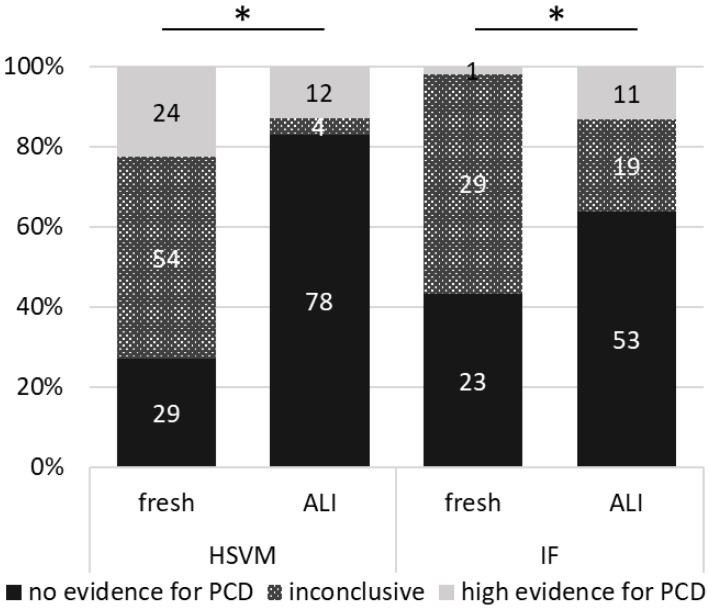
Differences in the assessments of ciliary motility via HSVM and presence of structural ciliary proteins via IF stainings in fresh and ALI samples. For HSVM fresh, we analyzed 107 samples in total, as re-brushings were performed for seven patients. Two of the samples could not be analyzed at all since no viable cells were found. We had a usable cell culture for 94 patients. For five patients we had no ALI and for one patient, the cell culture was not stable and lost all cilia before reaching maturity (at 28 days), and could not be analyzed. * Statistically significant difference between HSVM fresh versus ALI for “no evidence for PCD” (*p* < 0.0001), “inconclusive” (*p* < 0.0001) and “high evidence for PCD” (*p* = 0.04) and IF fresh versus ALI for “inconclusive” (*p* = 0.03). Differences between IF fresh versus ALI were not significant for “no evidence for PCD” (*p* = 0.18) and “high evidence for PCD” (*p* = 0.38). Data were compared using the Wilcoxon matched-pairs signed rank test (GraphPad Prism Version 9.0.2).

**Figure 4 diagnostics-11-01540-f004:**
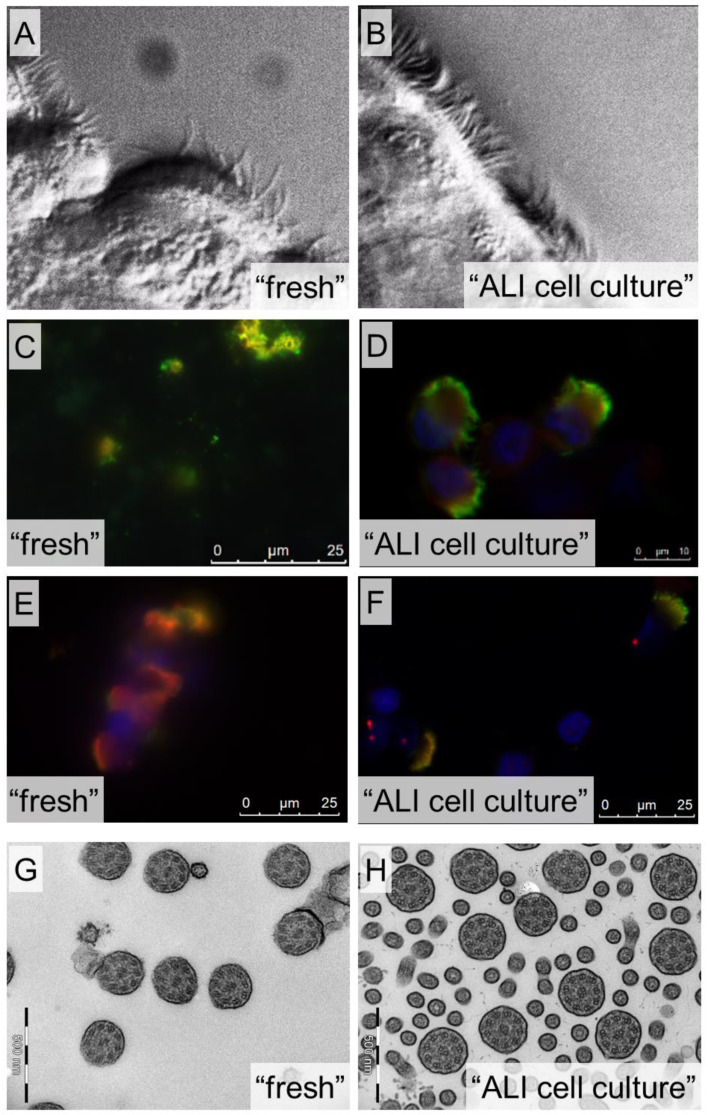
Examples of quality differences between fresh and ALI cell culture samples for HSVM (**A**,**B**), IF (**C**–**F**), and TEM (**G**,**H**). In **C**–**F**, the green staining shows the tubulin of the ciliary axoneme, the red staining shows the GAS8 (**C**,**D**) or DNAH5 (**E**,**F**) protein, and the blue staining the cell nuclei.

**Table 1 diagnostics-11-01540-t001:** Characteristics of the study population, summary of the referring centers, and overview of the diagnostic status and outcome. Percentages are related to the total number of cases in the respective column.

Characteristic of the Study Population	All Cases	PCD Cases	Non-PCD Cases
Mean age (Standard deviation), years	12.9 (18.0)	29.2 (26.4)	9.1 (12.8)
Median age (range), years	5.6 (0.04–68.9)	16.9 (1.4–68.9)	4.8 (0.04–61.7)
Sex (female/male), *n*	44/56	8/9^1^	36/44 ^1^
Age category ^2^ (children/adults), *n*	83/17	11/6	70/10
**Referring centers, *n* (%)**	**all cases**	**PCD cases**	**non-PCD cases**
Inselspital, Bern University Hospital (pediatrics)	54 (54%)	7 (41%)	45 (56%)
Inselspital, Bern University Hospital (adults)	2 (2%)	0 (0%)	1 (1%)
Cantonal Hospital Graubünden (pediatrics)	23 (23%)	3 (18%)	20 (25%)
Children’s Hospital Lucerne	4 (4%)	1 (6%)	3 (4%)
Fribourg Hospital (pediatrics)	4 (4%)	0 (0%)	4 (5%)
Lausanne University Hospital (pediatrics)	2 (2%)	0 (0%)	2 (3%)
Hospital Lindenhof (Pneumology, Bern)	4 (4%)	2 (12%)	2 (3%)
Others ^3^	7 (7%)	4 (24%)	3 (4%)
**Symptoms, *n* (%)**	**all cases**	**PCD cases**	**non-PCD cases**
Chronic productive cough	61 (61%)	9 (53%)	51 (64%)
Chronic rhinitis	40 (40%)	9 (53%)	29 (36%)
Chronic sinusitis	8 (8%)	1 (6%)	7 (9%)
Recurrent otitis media	18 (18%)	4 (24%)	13 (16%)
Recurrent respiratory infections	41 (41%)	2 (12%)	36 (45%)
Bronchiectasis	12 (12%)	3 (18%)	8 (10%)
Neonatal respiratory distress	8 (8%)	1 (6%)	7 (9%)
Situs anomalies	10 (10%)	5 (29%)	5 (6%)
Others ^4^	42 (42%)	9 (53%)	32 (40%)
**nNO^3^, *n* (%)**	**all cases**	**PCD cases**	**non-PCD cases**
Reduced (<77 nL/min)	25 (25%)	3 (18%)	21 (26%)
Borderline (<85 nL/min, >77 nL/min)	2 (2%)	11 (65%)	14 (18%)
Normal (>85 nL/min)	24 (24%)	0 (0%)	2 (3%)
Not performed ^5^	49 (49%)	3 (18%)	43 (54%)

^1^ Three cases were unsolved (*n* = 2 inconclusive, *n* = 1 diagnostics ongoing); thus, PCD and non-PCD cases do not sum up to the numbers of all cases. ^2^ Children = age younger than 18 years. ^3^ Hospitals with only one case, e.g., University Children’s Hospital Basel, University Hospital Basel (Clinic for Ear–Nose–Throat). ^4^ e.g., cardiological problems, obstructive sleep apnea syndrome, growth impairment, hearing problems, colonization with *Pseudomonas aeruginosa*, family history of PCD. ^5^ For some patients we do not have this information either because the referring center had no possibility to measure nNO levels or because the patient was too young to perform a nNO measurement.

**Table 2 diagnostics-11-01540-t002:** Diagnostic outcome of the first 100 patients referred to PCD-UNIBE.

Diagnostic Outcome	*n* (%)
no evidence for PCD	80 (80%)
high evidence for PCD (=diagnosis of PCD)	5 (5%)
PCD positive (=diagnosis of PCD)	12 (12%)
Inconclusive ^a^	3 (3%)

^a^ Currently inconclusively due to refusal for further investigations in two patients and still ongoing diagnostics in one patient.

**Table 3 diagnostics-11-01540-t003:** Overview of the different methods and whether the outcomes led to the diagnosis of PCD.

n	HSVM	IF	TEM	Genetics	Remark
8	+	+	-	+	all DNAH11 mutations
2	+	+	+	(not done)	no genetics performed
2	-	+	+	-	no mutations found
1	+	+	-	-	one pathologic variant in DNAH11
1	+	-	+	-	genetics still ongoing
1	+	-	(not done)	+	HYDIN mutation
1	-	**+**	-	-	No mutation found
1	-	-	(not done)	**+**	DNAH5 mutation, cell culture not successful
**Total: 17**	**13**	**14**	**5**	**10**	

*n* = number of patients, **written in bold** = the sole method leading to the diagnosis of PCD. + = the outcome of this method was leading to the diagnosis of PCD. - = the outcome of this method was neither “high evidence for PCD” nor “PCD positive” and, thus, was NOT leading to the diagnosis of PCD.

**Table 4 diagnostics-11-01540-t004:** Overview of the costs for the different methods. These costs only include running costs, such as consumables, fee for using microscopes and working hours, but no initial set up costs (e.g., purchase of equipment) and neither costs on the side from clinics or patients for the visits (of which more are needed if only fresh material is used).

Investigation/Method	Consumables and Fee for Equipment Use	Working Hours	Total
nasal brushing	EUR 6	EUR 70	EUR 76
cell culture	EUR 250	EUR 260	EUR 510
HSVM fresh	EUR 24	EUR 180	EUR 204
HSVM ALI	EUR 24	EUR 140	EUR 164
IF fresh ^1^	EUR 155	EUR 170	EUR 325
IF ALI ^1^	EUR 155	EUR 120	EUR 275
TEM fresh	EUR 500	EUR 1500	EUR 2000
TEM ALI	EUR 250	EUR 750	EUR 1000
Reporting/meetings/etc.	EUR 0	EUR 181	EUR 181
Genetics ^2^	EUR 200	EUR 2800	EUR 3000
all standard methods ^3^	EUR 459	EUR 770	EUR 1229
all methods ^4^	EUR 909	EUR 4501	EUR 5410

Remark about costs: PCD-UNIBE is based in Switzerland where salaries and prices for consumables are usually higher than in other European countries. Original costs were calculated in Swiss francs. For the conversion to Euros, an exchange rate of 1.1 CHF per one EUR was used. ^1^ These costs apply for the standard panel including DNAH5, GAS8, and RSPH9. Costs are higher if additional proteins are stained. ^2^ Average costs, costs vary depending on affected gene. ^3^ Standard methods include nasal brushing, cell culture, HSVM fresh, HSVM ALI, and IF ALI. ^4^ All methods include nasal brushing, cell culture, HSVM fresh and ALI, IF ALI, TEM ALI, and genetics.

## Data Availability

The data presented in this study are available upon request from the corresponding authors. The data are not publicly available due to privacy issues and data protection.

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
