# Peer review of "A Comprehensive Approach for the Diagnosis of Primary Ciliary Dyskinesia—Experiences from the First 100 Patients of the PCD-UNIBE Diagnostic Center"

_diagnostics, 2021, doi:10.3390/diagnostics11091540_

Round 1
Reviewer 1 Report
This work about the complex diagnosis of PCD, if not new, but it is presented in a very complete detailed way, it will certainly serve as a guide to anyone wanting to 1) set up a PCD comprehensive service or 2) to complement their own service with more sophisticated techniques and procedures. I think the way this work is prepared will contribute valorously for the PCD community.
Therefore, I only have some comments that I will list below:
I would suggest the authors to rephrase the bracketed sentence on 112-113: "(since for some 112 patients not the patients are referred to PCD-UNIBE, but only the nasal brushes are sent)." since it is not very clear;
On figure 4 I recommend the author to chance the IF image on the right (ALI) to one more comparable to the one on the left (fresh), and to find a region that do not represent single cells since the authors elongate during their discussion about the benefits of ALI producing strips of cells rather than single cells. Also, on the same image but on the TEM panel, I would prefer to see a low magnification of the epithelium showing the differences between ALI and fresh described in the text.
Table 2: minor recommendation, change the foot note 1 to a symbol since, on the same page there is already a footnote numbered as 1. nevertheless, I consider this Table 2 to be redundant and I would remove it.
Appendix A on TEM section: was the stage tilted to access a better plane of the cilia sections? if yes, consider mentioning it since is an procedures that can help in a clearer assessment of the section.
Author Response
Responses to reviewer 1:
This work about the complex diagnosis of PCD, if not new, but it is presented in a very complete detailed way, it will certainly serve as a guide to anyone wanting to 1) set up a PCD comprehensive service or 2) to complement their own service with more sophisticated techniques and procedures. I think the way this work is prepared will contribute valorously for the PCD community.
Response: We thank the reviewer for the valuable comments and the very positive feedback. Below are our point-to-point responses.
- I would suggest the authors to rephrase the bracketed sentence on 112-113: "(since for some 112 patients not the patients are referred to PCD-UNIBE, but only the nasal brushes are sent)." since it is not very clear;
Response: We rephrased the bracketed sentence as required. The new sentence is as follows:
“(since some patients are not physically present at PCD-UNIBE, but only their nasal brushes are sent to us)”
- On figure 4 I recommend the author to chance the IF image on the right (ALI) to one more comparable to the one on the left (fresh), and to find a region that do not represent single cells since the authors elongate during their discussion about the benefits of ALI producing strips of cells rather than single cells. Also, on the same image but on the TEM panel, I would prefer to see a low magnification of the epithelium showing the differences between ALI and fresh described in the text.
Response: We changed the ALI IF image to an image with more cells. However, we do usually not see stripes of cells on our IF slides as we dissociate the cell monolayer with Accutase before transferring them to the IF slides. Unfortunately, we do not have llow magnification TEM images. However, the overview on the ciliary lining and its continuity is much better visible on light microscopic sections (see panel A vs. B from fig. 4) Furthermore, the difference in TEM done on sections from ALI vs from fresh brushes changes primarily the effort until an accurate number of evaluable axonemal cross-sections is captured. This is sometimes visible in the mean density of cross-sections per field of view and the improved orthogonal orientation of large groups of cilia per field (as shown in fig. 4 E vs. F). However, fresh brushes may provide as suitable single cross-sections as ALI derived material but requiring much more photographs to get an acceptable evaluation.
- Table 2: minor recommendation, change the foot note 1 to a symbol since, on the same page there is already a footnote numbered as 1. nevertheless, I consider this Table 2 to be redundant and I would remove it.
Response: The footnote indicator on table 2 was adapted accordingly (from a “1” to a “a”). We agree that Table 2 is somehow redundant to the information in the text. However, it is the only table that allows a quick insight to the overall outcome of our 100 patients, and we would prefer to leave it. Nevertheless, in case this is asked from the editor, we can also remove it.
- Appendix A on TEM section: was the stage tilted to access a better plane of the cilia sections? if yes, consider mentioning it since is an procedures that can help in a clearer assessment of the section.
Response: Indeed minor tilting was applied to get better delineation of the axonemal structure. We added this important information to the text in Appendix A:
“One axis tilting of the specimen holder in the range of maximally 20° is sometimes used to increase the delineation of axonemal structures and dynein arms.”
Reviewer 2 Report
The paper is a comprehensive study using several methods. The authors examined 100 patients in detail by different methods: cell culture with analysis of ciliary motility, video-microscopy, immunofluorescence labelling, transmission electron microscopy and genetics analysis.
When reading the text for the first time a feeling of overloading with data is created. However, the schemes given by the authors greatly facilitate the understanding of the material presented.
The article can be accepted for publication after correcting the following remarks:
- I would recommend deleting incomplete results from the text of the article.
For example:
Line 265
“Genetic analysis was carried out in 31 cases, with one analysis still ongoing.”
Line 283
« Two patients refused further investigations and in one case the diagnostics is still ongoing.”
In my opinion, if the analysis is not completed and the result is not obtained, then I would consider such data as "not made analyzes".
- In figure 4 photographs C and D are given at different magnifications. It is difficult to compare them in this case. It is necessary to give both photographs at the same magnification, preferably choosing a higher magnification so that the reader can see the details. As such, photo 4C is not informative. Where are the nuclei of the cells in this photo?
- It is necessary to add immunofluorescent staining obtained with all other antibodies to illustrate the results.
- In Materials and Methods, all things are usually described in the past tense. This needs to be coordinated with the Editorial Board.
- When describing cell fixation, it is always necessary to indicate the concentration of the fixative and the buffer on which the fixative is prepared (pH and molarity of the buffer). The phrase "fixation with glutaraldehyde" is insufficient to describe.
For example:
Line 600: «1ml of GA is added to a 1.5ml Eppendorf tube.»
Footnote 8 after line 632: «For that, we add 1ml PFA to a 1.5ml tube, place the half membrane in it and let it at RT for 20min.»
- Line 707.
“TEM processing is performed as previously described (Nussbaumer et al, ERJ open, in revision).”
If the article has not yet been accepted for publication, then you cannot refer to it when describing the method.
Author Response
The paper is a comprehensive study using several methods. The authors examined 100 patients in detail by different methods: cell culture with analysis of ciliary motility, video-microscopy, immunofluorescence labelling, transmission electron microscopy and genetics analysis.
When reading the text for the first time a feeling of overloading with data is created. However, the schemes given by the authors greatly facilitate the understanding of the material presented.
The article can be accepted for publication after correcting the following remarks:
Response: We thank reviewer 2 for her/his valuable comments and the positive outcome.
- I would recommend deleting incomplete results from the text of the article. For example: Line 265 “Genetic analysis was carried out in 31 cases, with one analysis still ongoing.” Line 283 « Two patients refused further investigations and in one case the diagnostics is still ongoing.”
In my opinion, if the analysis is not completed and the result is not obtained, then I would consider such data as "not made analyzes".
Response: As stated in our manuscript, we report the authentic situation of our comprehensive center and try to include also inconclusive case, as this is reality in our center. It is not extremely seldom that patients refuse further investigations although interim results of our PCD diagnostics are not conclusive. In other cases, the current state of genetic characterization is not yet able to classify a defect, but the structural and functional results are suggestive. A result may need longer time to be available. Both situations are in our view part of the “daily business” of a PCD diagnostic facility and we think this should be reflected in this manuscript. However, we added a comment in the “Discussion” (paragraph “Limitations and strengths”, after line 400:
“Three cases remained inconclusive due to missing analytic data (2x refusal of further investigation, 1x ongoing analysis) but are consciously part of the first 100 patients referred to PCD-UNIBE. Such situations are an authentic part of the daily work in our PCD diagnostic center.”
If you are not satisfied with this way, of course we can remove the patients and refer only to the first 97 patients.
- In figure 4 photographs C and D are given at different magnifications. It is difficult to compare them in this case. It is necessary to give both photographs at the same magnification, preferably choosing a higher magnification so that the reader can see the details. As such, photo 4C is not informative. Where are the nuclei of the cells in this photo?
Response: The magnification of the images is the same (63x), however, the scale bar is different. While the fresh slides mostly shows debris and broken cells, the ALI shows nicely intact cells. However, as also asked by reviewer 1, we changed the image of the ALI slides to have a more similar image compared to the fresh one.
- It is necessary to add immunofluorescent staining obtained with all other antibodies to illustrate the results.
Response: In addition to the images of a GAS 8 staining, we added a fresh and an ALI image of DNAH5. We believe that this should be enough to show exemplary the differences between fresh and ALI samples and think it would be too much to show examples of every single antibody.
- In Materials and Methods, all things are usually described in the past tense. This needs to be coordinated with the Editorial Board.
Response: We described all protocols of our workflow in present tense as these are still performed. We described parts that are specific for this paper in past tense. However, we can change this if wished by the editor.
- When describing cell fixation, it is always necessary to indicate the concentration of the fixative and the buffer on which the fixative is prepared (pH and molarity of the buffer). The phrase "fixation with glutaraldehyde" is insufficient to describe.
For example: Line 600: «1ml of GA is added to a 1.5ml Eppendorf tube.»
Footnote 8 after line 632: «For that, we add 1ml PFA to a 1.5ml tube, place the half membrane in it and let it at RT for 20min.»
Response: We included all details about the used buffer in the parts of the specific methods. The details of the GA buffer are given in the TEM section and the details about PFA in the IF section. However, to help the reader finding the information we added a clearer hint to where the information can be found (underlined words are new).
For GA: “glutaraldehyde (GA, details of concentrations see below)”
For PFA: “paraformaldehyde (4% PFA, the same as used for fixation during IF, details see below)”
- Line 707. “TEM processing is performed as previously described (Nussbaumer et al, ERJ open, in revision).”
If the article has not yet been accepted for publication, then you cannot refer to it when describing the method.
Response: Meanwhile the article in ERS open is accepted and we changed the reference to “Nussbaumer et al, ERJ open, in press”.